# Methods to Prevent Future Severe Animal Welfare Problems Caused by COVID-19 in the Pork Industry

**DOI:** 10.3390/ani11030830

**Published:** 2021-03-16

**Authors:** Temple Grandin

**Affiliations:** Department of Animal Science, Colorado State University, Fort Collins, CO 80523, USA; cheryl.miller@colostate.edu

**Keywords:** COVID-19, pigs, animal welfare, euthanasia, supply chain

## Abstract

**Simple Summary:**

In the U.S., thousands of pigs had to be destroyed on the farms when illness caused by COVID-19 greatly reduced pork slaughter plant capacity. Some of the methods used to destroy pigs on the farms severely compromised animal welfare. Reliance on a few large slaughter plants created a fragile supply chain. Animal welfare auditing conducted by large meat buyers was also hindered by COVID-19. Many live in-person audits were stopped and replaced by a combination of stationary video cameras and live streamed videos from mobile phones. To insure high standards of animal welfare, video methods should never completely replace in-person visits.

**Abstract:**

In the U.S., the most severe animal welfare problems caused by COViD-19 were in the pork industry. Thousands of pigs had to be destroyed on the farm due to reduced slaughter capacity caused by ill workers. In the future, both short-term and long-term remedies will be needed. In the short-term, a portable electrocution unit that uses scientifically validated electrical parameters for inducing instantaneous unconsciousness, would be preferable to some of the poor killing methods. A second alternative would be converting the slaughter houses to carcass production. This would require fewer people to process the same number of pigs. The pandemic revealed the fragility of large centralized supply chains. A more distributed supply chain with smaller abattoirs would be more robust and less prone to disruption, but the cost of pork would be greater. Small abattoirs can coexist with large slaughter facilities if they process pigs for specialized premium markets such as high welfare pork. The pandemic also had a detrimental effect on animal welfare inspection and third party auditing programs run by large meat buyers. Most in-person audits in the slaughter plants were cancelled and audits were done by video. Video audits should never completely replace in-person audits.

## 1. Introduction

In the U.S., the most severe farm animal welfare problems due to COVID-19 were in the pork industry. Thousands of pigs had to be destroyed on the farms [1]. This was both a huge animal welfare and food waste issue. There were several reasons why the pork industry had more problems that severely compromised animal welfare compared to poultry or beef. When COVID-19 infected hundreds of slaughter plant employees, the large plants either shut down or ran at decreased capacity. In April and May 2020, plant closures and reduced staffing resulted in the pork slaughterhouses running at 40–45% capacity [2,3]. By the summer, the plants were back running at 90–95% capacity [4]. For a critical two month period in April and May 2020, thousands of pigs were at market ready weights with no place to process them. Since cattle are ruminants, it was much easier to slow down growth and wait for the plants to reopen [5]. They can be switched from a high grain to a high forage diet. Pigs are monogastrics with a digestive tract that is similar to humans. There are rations that can be used to slow down their growth but they are less effective [6]. Adding more fiber and reducing energy in the diet will increase the time for a pig to reach market weight. This may only provide four to seven days of extra time [6]. Poultry have a much shorter life cycle than pigs and it is easier to stop production by hatching fewer eggs.

Data from the Minnesota Pork Producers showed that 350,000 pigs were euthanized and 250,000 were sold at auctions, slaughtered in other facilities or given away [1]. Another variable that contributed to the huge number of pigs that had to be destroyed in the U.S. was the 2018 outbreak of African Swine Fever in China. This greatly reduced China’s pig population [7]. At the time that COVID-19 infected many employees in the U.S., large amounts of pork were being exported to China [7,8,9]. To satisfy a great increase in demand, U.S. producers increased their pig production. This clearly showed that the supply chains in these two countries were intertwined. A disease that killed a large numbers of pigs in one country was a stimulus for another country to increase pork exports. When COVID-19 either shut down or slowed the slaughter lines market, pigs were readily backed up on the farms. Some of the pigs that had to be disposed of were transported to a large commercial slaughter plant where their standard stunning equipment was used to kill them. This required only a small crew of people. All the carcasses were sent to either compost or landfill [2]. From a welfare perspective, this option was preferable to killing pigs on the farm.

On some farms, either a penetrating captive bolt or gunshot was used according to people who were on the farms. This is definitely an approved method that induces instantaneous unconsciousness [10,11,12]. One of the problems with a captive bolt for large numbers of animals is the guns will overheat. From the author’s experience in large slaughter plants, when a handheld cartridge fired captive bolt is used, multiple guns have to be rotated to prevent overheating. The use of a gunshot has worker safety issues due to dangerous projectiles. If firearms have to be used, the pigs should be shot outside. This will reduce ricochet hazards from bullets hitting a concrete floor. One of the problems with having to shoot thousands of animals is distress to the people. Several studies have shown that farmers and the people who have to kill many animals may get distressed and develop mental health problems [13,14,15]. People who spend their lives raising animals do not like killing them [16]. In the COVID-19 situation, healthy pigs had to be destroyed and all the meat was wasted. This would have made killing them even more stressful because raising the pigs no longer served a useful purpose.

Some of the methods used to depopulate the farms would not have been in compliance with the American Veterinary Medical Association (AVMA) as approved methods of euthanasia [10]. The AVMA does make a distinction between their euthanasia and their depopulation guideline [10,17]. There was a big controversy in the U.S. about the use of ventilation shutdown as a method of killing pigs on the farm. The reference provided in the AVMA depopulation guideline clearly showed that shutting off the ventilation systems with no additional interventions does not work [17,18]. When ventilation shutdown plus is used, the pit of a slatted floor barn is filled in and all fresh air inlets are blocked. Heat and humidity is added with a steamer and strict process control procedures are used to prevent scalding the pigs. The author watched a video of the interior of the barn and there was little behavioral reaction from the pigs. To do it correctly would require considerable engineering expertise. Research is also needed to determine the time of onset of unconsciousness. According to the depopulation guideline, this method should only be used in “constrained” circumstances. The use of this terminology does not provide clear guidance [17]. The author recommends that some examples of constrained circumstances where ventilation shutdown plus may be justified should be added to the AVMA depopulation guideline. One example would be a foreign animal disease. There were many critical articles in the news media about the use of ventilation shutdown plus causing suffering [19,20]. A recent review of the scientific literature on swine depopulation stated that “none of the published studies demonstrated an ideally reliable and safe way to induce rapid unconsciousness in large groups of pigs” [21]. This review missed a paper by Dutch researchers that was published in 1986 [22]. In this paper, Bert Lambooy described a high voltage electrical tunnel that pigs moved through on a moving conveyorized floor. It used 1000 volts and the pigs were killed when their heads hit an electrified curtain of chains. This is the only research study that has been published in the scientific literature [22].

## 2. Short-Term Easy to Implement Solutions to Reduce Future Severe Welfare Problems

The methods to reduce animal welfare problems discussed in this paper are based both on the scientific literature and practical experience the author has working with the livestock industry. Some of the areas I have worked in are, designing animal handling facilities, solving handling and stunning problems, conducting animal welfare audits, and training auditors. Other areas are writing welfare guidelines and serving on corporate animal welfare advisory committees. The emphasis in this next section is going to be on practical recommendations that would be reasonably easy to implement if a large numbers of pigs have to be depopulated on a farm.

### 2.1. Portable Electrocution Trailer Is a Viable Method for On-Farm Depopulation

Research studies on electrical stunning methods used in commercial slaughterhouses have shown that when a sufficient current is passed through the brain, pigs, cattle, and sheep will become instantly unconscious [23,24]. There have been two demonstrations that illustrate the potential of a portable electric method to be an economical and humane method for mass euthanasia of pigs on the farm. The demonstrations described below should be used as a starting point for the development of a scientifically verified portable electrical system. In 2011, the system researched by Bert Lambooy was demonstrated to the Canadian Swine Health Board at a Farm in Poland [25]. The system was installed inside a truck so it could be easily transported. There were some problems with getting a reliable electrode contact in the correct position. One of the problems was that pigs stepping onto the moving conveyorized floor were not restrained. They would be able to easily back out. In large commercial abattoirs that use electric stunning, the pigs are held in a conveyor restrainer. For either a manual or an automatic electric stunner, this holds the pig for more accurate placement of the electrodes. Large-scale existing electrical stunning systems have either a V-conveyor restrainer or a center track (belly) (monorail) conveyor system. These systems have been used commercially for many years.

Research has shown that a properly designed belly conveyor is a low stress way to restrain an animal [26]. To assess the stressfulness of restraint or problems with stunning, there should be a low percentage of pigs vocalizing with high pitched squeals [27] or cattle vocalizing [28]. These are easy-to-use outcome measures. Each animal scored as either vocalizing or silent.

A prototype trailer is being developed by Ruth Woiwode and Benny Mote at the Department of Animal Science, University of Nebraska. It consists of a V-conveyor restrainer from a commercial pork slaughter plant mounted on a trailer. It is equipped with electrode paddles, per the design in the expired patent by Grandin (1999) [29]. This design is simple to build and the author wants to make it very clear that the patent has expired. The design is now in the public domain. Anyone can use it. The pig’s forehead is in firm contact with the paddles before the pig’s front leg contacts the ground electrode. This provides the electrode position of forehead to upper foreleg. About fifty cull pigs, ranging from 220 (100 kg) to 600 (272 kg), have been successfully euthanized with the prototype. Since meat quality and prevention of petechial hemorrhages is not an issue, higher voltages and amperages can be used to insure death. Low frequency 50 cycle (Europe) or 60 cycle (North America) electric currents should be used. Low frequencies of 50–60 cycles are more effective for inducing both cardiac arrest and instant unconsciousness [30,31]. To insure instantaneous unconsciousness, an electric stunner must induce a grand mal epileptic seizure [32,33]. When really high voltages are used, such as 1000 volts in the Dutch apparatus, the seizure may not be visible because the spinal cord neurons are disrupted. The head electrode must never be applied to the neck because the current will bypass the brain [34]. For electrical safety, the unit can be housed in an enclosed trailer.

### 2.2. Slaughter Plant with Reduced Staff Can Produce Carcass Pork

A method that could be used to reduce both the number of pigs that had to be destroyed on the farm and the tremendous waste of food would be the production of either carcass pork or large cuts, such as loins, hams, shoulders, or bellies [35]. Before COVID-19 infected the U.S., there was at least one large pork slaughter plant that was exporting whole hog carcasses to China [36]. This clearly shows that this option is feasible. Producing these large cuts could probably be accomplished with less than half of the abattoir employees. When the meat cutting floor is shut down, the plant would still be able to run its slaughter line at maximum capacity. This does not require retooling of the plant, but to switch over quickly would require some advance planning. Portable conveyor equipment may be required to bypass the packaging equipment. The question that many people have asked is, “How would these carcasses or large pieces of pork be distributed and sold?”. There are many places that have industrial-size kitchens that could be used to cut the meat up. Some of the examples are hotels, military bases, college dining services, and prisons. Many communities in the U.S., such as cattle ranchers would know how to cut up a carcass. Limited numbers of truckloads of meat could be sold to them. This could be accomplished with high standards of food safety. The carcasses would all be inspected by FSIS/USDA (Food Safety Inspection Service United States Department of Agriculture) because the pigs would be slaughtered in the large slaughter plant that already has FSIS/USDA inspection. In the U.S. and many other countries, chilled carcasses or large primal cuts are shipped all the time. Primal cuts are often transported in large plastic-lined boxes (combos) that fit on a forklift pallet.

## 3. The Big Is Centralized Supply Chain Fragile?

Many people will say big is bad. The real problem is that big is fragile [35]. The author has visited many large centralized pork slaughter plants. They can have excellent standards for both animal welfare and food safety. However, when a large pork slaughter plant is suddenly shut down there is a tremendous disruption of the supply chain [2]. The advantage of a large centralized supply chain is that it is extremely efficient [37]. When it is working correctly, the meat can be produced at a much lower cost [37]. The meat industry is not the only industry that has been disrupted by either COVID-19, bad weather, or some other disruptive event. Pharmaceutical supply chains are also very concentrated [38]. A disruption may have an effect on obtaining common generic medications. Container ships is another area where there are concerns about big being fragile. The largest ships can transport the equivalent of 10,000 truckloads of freight [39]. If something goes wrong, delays could be greater compared to several shipments on smaller ships. Telecommunications and internet infrastructure can also be damaged by floods, fires, or deliberate attacks. In the U.S., there was a recent bombing of a building housing centralized telecommunications equipment [40]. It housed landline, emergency, cellular, and internet services. Emergency communication and other services were disrupted up to 159 km away [40].

## 4. A Long-Term Solution Is the Creation of a More Distributed Supply Chain

COVID-19 has made many business leaders and producers realize that it may be wise to have a more distributed supply chains for many products. There have been numerous articles in the U.S. livestock and meat industry trade press about the need for more small slaughter plants [41,42]. Unfortunately, some of these articles are in livestock industry publications that are not readily available online. The demand for either modular or mobile slaughter facilities has greatly increased [41,42]. Consumers want more local food and producers need more small processing facilities [43]. In 2020, three or four groups of cattle producers have either started construction or have already built small and medium-sized plants [44,45]. These prefabricated units can also be expanded as the business increases. The units enable a group of pork or beef producers to more economically get a meat business started. There are two types of facilities for constructing smaller slaughter plants. They are the small prefabricated modular units and larger facilities that are built on site. The modular units can process a few hundred animals per week. The large medium-sized new facilities could handle 100–900 animals per day [46]. This is still small compared to a large U.S. plant that processes 20,000 pigs or 5000 cattle per day. To achieve these high numbers, they work two shifts each day.

It is easier for the cattle ranchers to adopt a more distributed supply chain than large modern swine finishers that have large numbers of pigs reaching market weight each day. For the swine industry, the most practical emergency option would be processing whole hog carcasses. Outlets for the carcasses need to be determined in advance. One possibility would be to use the services of meat science students and volunteers to cut them up for distribution to emergency food programs. People were going hungry during the COVID-19 pandemic and the meat from these animals could be used in these programs.

### Small Slaughter Houses Have to Have a Specialized Market to Be Profitable

To be profitable, small slaughter plants cannot compete on a cost basis with the largest slaughter plants. They need a specialized market where they can charge a higher price for a premium product. Some of the niche markets are grass-fed beef, high welfare outdoor pigs, pork produced on family farms, special sustainable practices, or a specific breed of animal [47,48]. Another niche is ethical local meat. In the U.S., Niman Ranch is a very successful niche market of high welfare pigs [49]. Many consumers are also concerned about the number of people that got sick and died in the large plants. This welfare of people is also an issue in the minds of consumers.

The author has been in the U.S. livestock industry for almost fifty years. In the 1980s and 1990s, she observed the sad fate of many medium-sized slaughter plants when they attempted to directly compete for the same customers with the larger plants. Plants in California, Colorado, Arizona, and Texas went out of business because they could not achieve the low per animal costs of the huge plants. There is a tradeoff. A centralized huge supplier is really economical but fragile when it is disrupted by a pandemic or storms. A more distributed supply chain is more robust, but the products will be more expensive. In the U.S. a combination of COVID-19, forest fires, and storms that cause severe flooding have made many consumers more concerned about their food security [50]. This may motivate more consumers to buy local.

## 5. Effect of COVID-19 on Animal Welfare Audits Conducted by Large Meat Buyers

For the last twenty years, large buyers of meat in the U.S. have been inspecting and auditing animal welfare at their suppliers. Large retailers are increasingly putting an increasing emphasis on the importance of regular welfare audits [51]. The author assisted in the development of some of the first audits of large U.S. slaughter houses [52]. The audits were started in 1999 and they resulted in huge reductions in electric prod use, and improvements of both handling and stunning. Captive bolt stunning was greatly improved by better stunner maintenance [53]. Handling of both cattle and pigs was also improved by repairs of handling equipment, installation of non-slip flooring, employee training, and other simple changes such as illuminating the entrance of a restrainer to facilitate animal entry [53]. In the U.S., the largest improvements occurred when large meat buyers demanded them [52]. Over the years, more and more buyers have started inspections and industry organizations have responded by both writing guidelines and starting training programs.

The next step in the early 2000s was the formation of PAACO (Professional Animal Auditor Certification Organization) [54]. PAACO is a consortium of meat buyers, academics, professional, industry, and veterinary organizations. Its purpose is to provide training and certification of animal welfare auditors. Another function is reviewing animal welfare guidelines that are written by livestock industry associations. The author is a PAACO instructor on animal welfare at slaughter. When COVID-19 stopped almost all the business travel in the U.S. PAACO auditor training was instantly converted to virtual on-line. The author has participated in these virtual programs. The two slaughter plant visits, which were originally part of these classes, were cancelled and replaced with video tours. This was the definite downside of switching to digital. There were some advantages. Previously, a typical PAACO animal welfare auditor training class for either slaughter or welfare on the farm had about 20 students. The classes were kept small to facilitate in-person training at farms and slaughter plants. When the training classes were switched to digital, the size of the classes tripled. More students enrolled from other countries. Online classes made animal welfare training available for more people. COVID-19 problems in the U.S. have kept PAACO classes online through the time of writing this article in January 2021.

In April and May 2020, third party independent welfare audits and audits by meat buyers were forced to convert to online virtual [55]. The large slaughter plants banned outside visitors to prevent the spread of COVID-19. Some people in the meat industry have told the author that welfare audits of farms and slaughter plants can be kept entirely virtual. The author does not agree with this. From her extensive experiences with being an animal welfare auditor, she has learned that there are too many ways to cheat. Some cheating methods that the author has personally observed are falsified scores where they were “too good”, electric prodding of cattle just outside the view of the video camera and fake electrical meters on a pig stunner. This was done to mask the use of low electrical amperages that were not effective. It is the author’s opinion that the audits could be made partially digital. This would be especially true for a farm or slaughter plant that an auditor has visited many times. The auditor could have a plant or farm employee walk around with a smart phone and livestream video of the parts that needed to be assessed. For an initial visit, an in-person visit would be essential to help prevent either livestock producers or slaughter managers from hiding hidden areas. Since 2008, a commercial auditing company in the U.S. has been using remotely viewed cameras installed in the abattoirs to monitor handling and stunning [56]. These cameras are connected with a hard wired internet connection. Due to COVID-19, some plant managers have now become more open to allowing buyers and customers access these video feeds.

When it works, the high quality of some of the video is amazing. The author has had the opportunity to do some consultation on both pig handling and determining if an animal was unconscious over an internet connected video link. It worked surprisingly well. Videos of pigs moving through a handling facility showed both really good low stress handling of pigs and poorer methods when the number of pigs moved at one time was too large. These videos would also be really useful for training employees. The downside is that in some rural areas of North America, internet service is poor and live streaming video from either a slaughter plant or farm either works intermittently or completely fails. This is especially a problem when auditing is done by having a plant employee walk through a plant with a mobile phone on cellular service. The video will often stop due to either a poor cellular signal or failure of the cellular signal to penetrate areas of the plant constructed from thick concrete or steel.

The author would be really concerned about welfare if there was an attempt to replace all in-person auditor visits with video. This would be especially a problem for welfare auditors and inspectors who have had little or no experience out on farms or in the plants. COVID-19 has forced both slaughter plant and farm management to become more cooperative about having video cameras in their facilities. The author predicts that the number of in-person audits could be reduced by the use of remote video. Total replacement of in-person visits would be a grave mistake.

## 6. Long-Range Thoughts on Food Supply Chains

There is a tendency for networks to form hubs, whether they are in Gingko trees, ferns, airline hub airports, or supply chains for the distribution of goods [57]. Ruth DeFries, a professor of ecology at Columbia University, explains that primitive plants, such as the Gingko, rely on a system of veins that depend on the minimum number veins to supply water to the leaves [56]. This system is fragile and the supply of water and nutrients can be easily cut off when the leaf is damaged. Modern plants have a “loopy network”, which has more redundancy because there is more than one pathway through the vein network. Modern plants evolved this more expensive network with extra veins to make the leaves less vulnerable to vein damage. In the airline industry, a hub and spoke system is efficient, but when a snowstorm disables a major airport, a large part of the entire system will have delays. We should learn from the evolution of plants and develop more distributed food supply chains. A recent editorial in the journal Nature states that the majority of scientific research is not relevant to insure food security to small farmers [58]. The editor concludes that research is needed to support small farmers. There is also a need to support both small farmers and small processing facilities to create a robust food production network. This principle applies to all foods.

Willy C. Shih wrote in the Harvard Business Review that to make supply chains for any product more robust requires either a diversity of sources or warehousing of key components [59,60]. Pigs are not electronics or industrial components that can be stored. The only sensible solution to prevent a repeat of an animal welfare and food waste disaster is a more diversified supply chain.

## 7. Conclusions

It is essential to develop programs so that large numbers of healthy pigs will not have to be destroyed on the farm due to a loss of slaughter capacity. If large numbers of pigs have to be euthanized on the farm, a portable electrical stunning system may be the best option. It can maintain the same high welfare standards that are required for electric stunning in a slaughter plant. From both a sustainability and animal welfare standpoint COVID-19 was a disaster for the pork industry. One solution is to develop a less centralized, more diversified options for pig slaughter and processing. The U.S. beef and poultry industries were less affected because it is easier to slow down the growth of cattle and chickens have a shorter life cycle. COVID-19 also revealed how international supply chains are dependent on each other.

## Data Availability

There is no data associated with this paper.

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
