# Peer review of "Methods to Prevent Future Severe Animal Welfare Problems Caused by COVID-19 in the Pork Industry"

_animals, 2021, doi:10.3390/ani11030830_

Round 1

Reviewer 1 Report

Most of the reference links provided did not work due to spelling mistakes in the link code. 

I have provided the correct link for few references as an example (attached document).

Please check all the weblinks whether they are working correctly so that the readers can understand the article.

Author Response

Reviewer 1

Line 32 – Corrected reference

Line 48-50 – Remove the sentence about the interview with Paul Yeske and replaced it with the Minnesota Pork Producers as the source of the euthanasia numbers. The reference was changed to the pigsite.

Line 61 – Corrected link

Line 63 – The sentence states that the pigs were shot with either a captive bolt or gunshot on the farm.  There is  no statement about performing it inside a barn.  Information on problems with captive bolts overheating and dangers of firearms have been added.

Line 82 – Added at the end of the sentence “causing suffering”

Corrected the AVMA link

Corrected the Bird reference

Reviewer 2 Report

The paper describe a very important topic in a professional way. I completely agree with the analysis of the issue, and thank the author for the well written, clear and really current description.

Only few minimal comments.

Author Response

Reviewer 2

  • Aligned the term and changed the numbering.
  • Reviewer 2 stated that “The paper described a very important topic in a professional way. I completely agree with the analysis of the issue, and thank the author for a well written, clear and really current description.” Reviewer 2 had no other revisions.

Reviewer 3 Report

Overall this is an important topic for discussion.

Line 63- Other constraints of these methods should be discussed. For captive bolt gun, with large numbers of animals, many guns are needed as they heat up and become unreliable so after a certain number of animals the gun needs to be traded out for a different gun. For gun shot there are worker safety issues with dangerous projectiles.  These are in addition to the mental health aspects which are not the only reasons these methods are not ideal.

Line 75- A better delineation could be made between VSD and VSD plus. With VSD plus the air space is restricted (by filling in the pits and blocking all fresh air inlets) and heat and humidity added. More study is needed on VSD plus as a method for depopulation as we don't have a good understanding of the time to unconsciousness when VSD plus is used. A more nuanced discussion of this method here and what more we need to understand about it would be warranted. The author's opinion is valued by many and too harsh a stance on this method could limit its study despite the fact that it could be an efficient and humane method if performed correctly. 

Line 81- Constrained circumstances were real during covid 19 and included the lack of available CO2 and the the constraints mentioned above for gun shot and captive bolt. More clarity as to how the language "constrained circumstances" could be improved, without limiting the options and potentially decreasing the health of the entire US swine herd in the event of an FAD, would be helpful. 

In general there is a lack of good research on portable electric methods. This should be better conveyed to the reader. It may well be a good option but we have one study from 1986 that was not done on the typical type of farm that was impacted here and doesn't examine important outcomes in a standardized way. We also have a report in Canada that never made it into the peer reviewed literature. This paper makes the case that it is the ideal method but this reviewer feels that is too strong a stance based on the quality of information we have about the method. 

Line 151 states that truckloads of large pieces of pork could easily be sold. This is an oversimplification of the problem. During Covid 19, much of the pork that had been fabricated into larger pieces ended up sitting in cold storage because the food service market that usually creates demand for these products was decimated by the pandemic. It should be discussed in a more nuanced way since the idea that this market exists and is easily accessible is contrary to the US experience during 2020 and the constraints of Covid 19.  

Section 4- Lacks a discussion about the impact on pigs- Would such a decentralized system have an impact on pig welfare? Who would inspect all these facilities with FSIS already understaffed? Would the transport distance change? The type of transport? 

There is also a mismatch between the modern swine finisher and these small to medium sized processors. The suggestion to decentralize and capitalize on niche markets is much more complicated than what is suggested. If all the pigs on a large modern finisher facility reach 280 pounds over the course of only a few weeks, how would such a small/medium sized facility cope with processing that number of pigs? There is also the issue of niche markets requiring different production practices from conception to slaughter and so how would such a processing system mesh with the current system and help with the fragility the author describes inherent in the centralized system? The reader would benefit from understanding how the author sees this decentralized system altering how pigs are raised on the farm, from sow farm to finisher, in regular commercial production systems or if it would exist in tandem and somehow help in an emergency.

Author Response

Reviewer 3

Line 63 – Addressed the issue of the problem of the captive bolts overheating and the dangers of firearms.

Line 75 – Added a clearer description of ventilation shutdown plus and added that research is needed to determine the time of onset of unconsciousness.

Line 81 – Added that ventilation shutdown plus may be justified for a foreign animal disease.  After the paragraph about the Dutch research a sentence was added to indicate that this was the only published scientific research on electrical methods. In the section on the portable electrocution unit, I added that these demonstrations should be used as a starting point for development of a scientifically verified electrical system.

Line 151 – Clarified that the FSIS/USDA inspection would be done in the large plant where the pigs were slaughtered.  This plant would already have the inspectors.

Section 4 – At the end of the introduction information was added that it would be easier for cattle ranchers to create a more distributed supply chain.  Since this would be very difficult for a large modern finishing operation the most practical option would be processing whole carcasses.

Round 2

Reviewer 3 Report

Line 69- "danger" to "dangerous"

Line 168- Suggesting that immigrants know how to cut up carcasses seems like an overbroad and biased generalization that should not be published.

Line 169- I still object to the word "easily". Truckloads of pork is tons of meat. It would be hard for communities like those described to absorb the volume of pork that would be created in these scenarios, especially on an ongoing basis. 

Author Response

Line by Line Responses

Reviewer 3

Line 69 – Danger has been replaced with dangerous

Lines 219-221 – The words immigrant has been removed.  It now reads: Many communities in the U.S., such as cattle ranchers, would know how to cut up a carcass.  Limited numbers of truckloads could be sold to them.  The word easily was removed and it has now been made clear that only a limited number of truckloads could be sold to them.

Temple Grandin, Professor

Department of Animal Science